

# Behavior and biocompatibility of rabbit bone marrow mesenchymal stem cells with bacterial cellulose membrane

Marcello de Alencar Silva[1,*], Yulla Klinger de Carvalho Leite[1], Camila Ernanda Sousa de Carvalho[1], Matheus Levi Tajra Feitosa[1], Michel Muálem de Moraes Alves[2], Fernando Aécio de Amorim Carvalho[2], Bartolomeu Cruz Viana Neto[3], Maria Angélica Miglino[4], Angela Faustino Jozala[5,*] and Maria Acelina Martins de Carvalho[1,*]

[1] Integrated Nucleus of Morphology and Stem Cell Research, Federal University of Piauí, Teresina, Piauí, Brazil
[2] Antileishmania Activities Laboratory, Federal University of Piauí, Teresina, Piauí, Brazil
[3] Department of Physics/Advanced Microscopy Multiuser Laboratory/Laboratory of Physics Material, Federal University of Piauí, Teresina, Piauí, Brazil
[4] Departament of Surgery, Faculty of Veterinary Medicine and Animal Science, University of São Paulo, São Paulo, Brazil
[5] Laboratory of Industrial Microbiology and Fermentation Process, University of Sorocaba, Sorocaba, São Paulo, Brazil
* These authors contributed equally to this work.

Corresponding author
Maria Acelina Martins de Carvalho, mcelina@ufpi.edu.br

## ABSTRACT

**Background**. Tissue engineering has been shown to exhibit great potential for the creation of biomaterials capable of developing into functional tissues. Cellular expansion and integration depends on the quality and surface-determinant factors of the scaffold, which are required for successful biological implants. The objective of this research was to characterize and evaluate the *in vitro* characteristics of rabbit bone marrow mesenchymal stem cells (BM-MSCs) associated with a bacterial cellulose membrane (BCM). We assessed the adhesion, expansion, and integration of the biomaterial as well as its ability to induce macrophage activation. Finally, we evaluated the cytotoxicity and toxicity of the BCM.

**Methods**. Samples of rabbit bone marrow were collected. Mesenchymal stem cells were isolated from medullary aspirates to establish fibroblast colony-forming unit assay. Osteogenic, chondrogenic, and adipogenic differentiation was performed. Integration with the BCM was assessed by scanning electron microscopy at 1, 7, and 14 days. Cytotoxicity was assessed via the production of nitric oxide, and BCM toxicity was assessed with the MTT assay; phagocytic activity was also determined.

**Results**. The fibroblastoid colony-forming unit (CFU-F) assay showed cells with a fibroblastoid morphology organized into colonies, and distributed across the culture area surface. In the growth curve, two distinct phases, lag and log phase, were observed at 15 days. Multipotentiality of the cells was evident after induction of osteogenic, chondrogenic, and adipogenic lineages. Regarding the BM-MSCs' bioelectrical integration with the BCM, BM-MSCs were anchored in the BCM in the first 24 h. On day 7 of culture, the cytoplasm was scattered, and on day 14, the cells were fully integrated with the biomaterial. We also observed significant macrophage activation; analysis of the MTT assay and the concentration of nitric oxide revealed no cytotoxicity of the biomaterial.

**Conclusion**. The BCM allowed the expansion and biointegration of bone marrow progenitor cells with a stable cytotoxic profile, thus presenting itself as a biomaterial with potential for tissue engineering.

## INTRODUCTION

Researchers have been studying bone marrow mesenchymal stem cells (BM-MSCs) for their applicability in regenerative medicine, and for improving current methodologies (*DiMarino, Caplan & Bonfield, 2013*; *Wei et al., 2013*; *Kobolak et al., 2016*; *Li et al., 2016*). BM-MSCs are widely used in clinical and therapeutic use due to several factors: they are easily accessible; it is possible to achieve the necessary volume of cells in a short time, through culture replication; they allow autologous use or the treatment of several patients with a single sample, since the expression of HLA antigens is poor; they can be used without the need for HLA typing, making them ready for use in any patient. Even after being frozen, they preserve their characteristics, which allows the creation of bio-banks (*Wabik & Jones, 2015*).

The use of mesenchymal stem cells (MSCs) has shown promise in the field of regenerative medicine. Studies have investigated the use of MSCs in cardiovascular events (*Castellanos et al., 2016*), immunological dysfunctions (*Kaplan, Youd & Lodie, 2011*; *Zhao, Ren & Han, 2016*), bone repair (*Emmet et al., 2016*), cartilaginous and intervertebral discs (*Blanquer, Grijpma & Poot, 2015*), tendinosis (*Peach et al., 2017*), and hematological malignancies (*Wang, Qu & Zhao, 2012*), among others (*Schnitzler et al., 2016*; *Squillaro, Peluso & Galderisi, 2016*). Tissue engineering is a promising multidisciplinary field that involves the development of materials or devices capable of specific interactions within biological tissues (*Langer & Vacanti, 2016*). Advances in research have demonstrated biocompatibility between stem cells and biopolymers in the development of *in vitro* tissues capable of repairing injured areas (*Lima et al., 2017*; *Park et al., 2017*; *Weinstein-Oppenheimer et al., 2017*).

Several biomaterials with different physicochemical and mechanical properties have been developed, with biomedical purposes including tissue regeneration, drug delivery systems, new vascular grafts, or *in vitro* and *in vivo* tissue engineering supports (*Lin et al., 2013*; *Xi et al., 2013*; *Soheilmoghaddam et al., 2014*; *Zulkifli et al., 2014*; *Kim & Kim, 2015*; *Pires, Bierhalz & Moraes, 2015*; *Urbina et al., 2016*).

The scaffold surface can generate cellular responses which can affect adhesion, proliferation, migration, biointegration, and cellular function (*Abbott & Kaplan, 2016*). This interaction is especially important to define the degree of rejection of medical implants (*Achatz et al., 2016*).

Bacterial cellulose is an extracellular polysaccharide secreted primarily by *Gluconacetobacter xylinus*, an aerobic, Gram-negative, and chemoheterotrophic bacterium that can be grown in liquid medium from various sources of carbon and nitrogen, and basically uses glucose as the substrate. In culture medium, this microorganism produces very

PeerJ ________________________________________

fine fibers that intertwine, forming a film with a nanofibrillar structure (*Moosavi-Nasab & Yoursefi, 2011*; *Li et al., 2012*; *Panesar et al., 2012*). Nanofibrils of length from 20 to 100 nm intertwine, forming a three-dimensional network, resulting in a high degree of hydrophilicity (*Jozala et al., 2015*; *Rajwade, Paknikar & Kumbhar, 2015*), water retention capacity, and porosity, which allows selective permeability, adhesion of cell culture, and diffusion of the culture medium (*Cavka et al., 2013*; *Zepon et al., 2013*; *Ashok et al., 2015*; *Kirdponpattara et al., 2015*).

Many studies have used the bacterial cell membrane *in vitro*, in preclinical studies investigating drug, hormone, and protein release systems, artificial skin (*Fu, Zhang & Yang, 2013*), cartilage (*Cruz et al., 2016*), menisci (*Achatz et al., 2016*), intervertebral discs (*Fávaro et al., 2016*), valvular prostheses, artificial corneas, and the urethra (*Rajwade, Paknikar & Kumbhar, 2015*). However, it will be necessary to improve our knowledge of bacterial cellulose membrane (BCM) biointegration and biodegradation, especially with respect to BM-MSCs.

This purpose of this study was to characterize and evaluate rabbit BM-MSC behavior *in vitro* when associated with a BCM, by analyzing adhesion, expansion, and cellular integration with the biomaterial, as well as the ability to induce macrophage activation. BCM cytotoxicity and toxicity were also evaluated.

## MATERIAL AND METHODS

### Study design

Bone marrow samples were collected from three adult rabbits and used for isolation and cryopreservation of MSC. A *Mus musculus* mouse was used as a source of peritoneal macrophages. To determine cellular viability, Trypan Blue staining and growth curve analysis were performed. For the fibroblastoid colony-forming unit assay, cells collected from the bone marrow (BM) cultured in 24-well plates at passage 6 were used. Chondrogenic, osteogenic, and adipogenic induction were used to assess the potential for differentiation into mesenchymal lineages. To verify BM-MSC biointegration with the BCM, inverted light microscopy and scanning electron microscopy (SEM) were used to analyze the phagocytic capacity, toxicity, and cytotoxicity of the BCM. This study was performed in strict accordance with the recommendations of the Guide for the Care and Use of Laboratory Animals of the National Institutes of Health. The protocol was approved by the Ethics Committee on the Use of Animals of the Federal University of Piauí (permit number: 268/16).

### Anesthetic protocol for bone marrow collection

After solid anesthetic fasting of 4 h, and 2 h of liquids, the rabbit was chemically restrained with a combination of 35 mg/kg of ketamine hydrochloride and 3 mg/kg of midazolam maleate. Trichotomy of the major trochanter region was performed, followed by antisepsis by femoral puncture with a 5 mL syringe; a heparinized 40 × 12 mm needle was used to obtain a BM sample. For antibacterial prophylaxis, 10 mg/kg of enrofloxacin was given twice daily for 7 days, and 25 mg/kg of sodium dipyrone plus 3 mg/kg of tramadol was administered twice daily for 3 days for pain control (*Ninu et al., 2017*).

## BM-MSC isolation, cultivation, and expansion

The methodology presented was adapted from *Argôlo Neto et al. (2016)*. Medullary aspirate (1.5 mL) was diluted in phosphate-buffered saline (PBS) at a ratio of 1:1 in 15 mL conical tubes. The resulting contents were filtered through 100 $\mu$m mesh, deposited in a 15 mL conical tube containing Ficoll Histopaque at a ratio of 1:1 (Ficoll:BM), and centrifuged at 2,000 rpm for 30 min at 20 °C to separate the cellular constituents by density gradient. The whitish halo, rich in mononuclear cells, was aspirated with an automatic pipettor (Houston; Swiftpro, HTL Lab solution, Warsaw, Poland), immediately diluted in sterile PBS with 1% antibiotic (100 U/mL penicillin and 100 $\mu$g/mL streptomycin) for cell lavage, and re-centrifuged at 1,500 rpm for 10 min at 20 °C. BM samples were resuspended in complete Dulbecco's modified Eagle's medium (DMEM) containing 3.7 g/L sodium bicarbonate and 10–15 mM HEPES (Invitrogen, no. 15630080; Invitrogen, Carlsbad, CA, USA), pH 7.5, 15% fetal bovine serum (Invitrogen), 1% penicillin–streptomycin, 1% L-glutamine (Invitrogen), and 1% non-essential amino acids (Sigma, St. Louis, MO, USA), and cell viability was assessed. For this purpose, a 50 $\mu$L aliquot of each sample was diluted in 50 $\mu$L 0.2% Trypan Blue dye, and mixed in a sterilized glass vial for cell counting in a Neubauer chamber.

Cells were seeded in a six-well cell culture plate (TPP) at a density of $10^6$ cells/well in 2.0 mL of low-glucose DMEM, and kept in an incubator (Thermo Scientific Series II Water Jacket; Thermo Scientific, Waltham, MA, USA) at 37 °C in 5% $CO_2$ and 95% humidity. The wells were washed twice every 3 days with PBS solution containing 1% antibiotic (100 U/mL penicillin and 100 $\mu$g/mL streptomycin), followed by exchange of the culture medium until the cultures reached 80% confluency. Subsequently, the wells were subjected to trypsinization with 2.0 mL 1× trypsin (Invitrogen, no. 25200-114, 10× Trypsin–EDTA solution), and incubated at 37 °C for 5 min. Following this, trypsin was inactivated with the addition of 4.0 mL low-glucose DMEM. The solution was transferred to a 15 mL conical bottom tube, and centrifuged (FANEM refrigerated Cytocentrifuge MOD.280R Excelsa 4) at 20 °C and 1,500 rpm for 10 min.

The supernatant was discarded, the pellet was resuspended in 1.0 mL of DMEM, and a new cell count was performed. The cells in suspension were used for expansion. To do this, $10^6$ cells/mL in 25 cm$^2$ tissue culture bottles with 3.0 mL of supplemented DMEM were incubated at 37 °C in 5% $CO_2$ and 95% humidity. The cultures were expanded and photographed with an inverted phase-contrast microscope (COLEMAN NIB-100), and peaked with twice the original area; cell concentration was verified at each passage.

## Fibroblastoid colony-forming unit assay

After plating $1 \times 10^4$ cells/mL of the BM-MSC rabbit fraction in 24-well plates, plates were observed daily to monitor the establishment of colonies with more than 30 cells. Cells were then fixed with 4% paraformaldehyde for 30 min, and stained with Giemsa for 10 min at room temperature. Any excess stain was washed away with distilled water. The colonies were observed and macroscopically counted on the 24-well plates (*Paramasivam et al., 2016*).

## Cell viability

Cell count, which determines concentration and viability, was performed using the Trypan Blue exclusion method. After mixing 30 μL of the cell suspension with 30 μL Trypan Blue solution (50 μL of 4.25% sodium chloride in 200 μL of Trypan Blue), a 10 μL aliquot was observed in a Neubauer chamber under an optical microscope (10× objective). The BM-MSC growth curve was performed in duplicate by plating $1 \times 10^4$ cells/mL in five six-well plates, and counting two wells every 24 h over the course of 15 days. The culture medium of the plates was changed every 3 days to maintain nutrient availability (*Sangeetha et al., 2017*).

## Cell differentiation

For cell differentiation assays, we use the protocols provided by Stem Pro®. Analysis of cell differentiation potential was performed with sixth-passage BM-MSCs cryopreserved in liquid nitrogen for 12 months. They were thawed and grown in 25 cm² bottles for cell expansion until 80% confluency was reached. Cultures were then trypsinized and seeded at the concentration according to the manufacturer's instructions, for chondrogenic, osteogenic, and adipogenic differentiation.

For chondrogenic differentiation, $3 \times 10^5$ cells per well were seeded in a 96-well plate. After 48 h, formation of spheroid bodies was observed, and the culture medium was replaced with that from a Stem Pro® Chondrogenesis Differentiation Kit. Exchange of the medium was performed every 3 days during a 21-day period. Analysis was performed with histological sections stained with Alcian Blue.

For osteogenic differentiation, $6 \times 10^4$ cells were seeded in a 24-well plate. Initially, the supplemented culture medium was removed and replaced with the osteogenic induction medium, and changed every 3 days during a 21-day period. During this period, morphological characteristics of the cells were evaluated. After osteogenic differentiation, cells were stained with Alizarin Red, which identifies the calcium-rich extracellular matrix, and is characteristic of the presence of osteoblasts. To do this, the cell monolayer was washed with PBS, and fixed with 10% alkaline phosphatase (AP) for 30 min at room temperature. The AP was then removed, the cell monolayer was washed with distilled water, and Alizarin Red was added for 5 min. Subsequently, the dye was removed, and five washes were performed with distilled water; the calcium-rich extracellular matrix and the amount of calcium deposits were recorded with an inverted light microscope.

For adipogenic differentiation, $2 \times 10^4$ cells per well were seeded in a 24-well plate, and Stem Pro Adipogenesis Differentiation Kit induction medium was added once the cells reached 80% confluency. The culture medium was exchanged every 3 days over a period of 10 days. Once differentiation occurred, the culture was stained with Oil Red to visualize lipid vacuoles.

## BM-MSC biointegration with the BCM

The bacterial cellulose membrane used in this study was developed in partnership with the pharmacy department of Sorocaba University - UNISO (Sorocaba, São Paulo-Brazil). MCB was obtained from the culture of G. xylinus ATCC 53582 prepared using 100 ml of

the Hestrin and Schramm medium at 30 °C for 48 h under agitation of 150 rpm. After that, $10^6$ cells/mL-1 were withdrawn from the culture medium. For the production of MCB, 24 well plates were used. Each well was filled with 1mL of inoculated culture medium. Plates were maintained at 30 °C in a static culture for 0, 24, 48, 72 and 96 h. MSCs were seeded onto the membrane (*Jozala et al., 2015*).

To study BM-MSC expansion and biointegration with the BCM, $2 \times 10^4$ cells were cultured in 12-well plates on BCM for three distinct periods (1, 7, and 14 days). The BM-MSCs were fixed to the BCM using 3% glutaraldehyde, washed once with PBS, and dehydrated by slow water exchange using a series of ethanol dilutions (30%, 55%, 70%, 88%, 96%, and 100%) for 20 min at each concentration. For analysis by SEM (FEI Quanta FEG 250), samples were fixed to the stub with double-sided carbon tape, placed in a dehumidifier for 2 h, and metalized with gold.

## Phagocytic activation

Phagocytic activity was assessed by collecting resident macrophages from the mouse peritoneum. The animal was euthanized by cervical dislocation after being reassured and sedated by intraperitoneal injection of a combination of xylazine hydrochloride and ketamine hydrochloride (10 and 80 mg/kg body weight, respectively). Macrophage removal was performed in a laminar flow hood with the animal affixed in the dorsal decubitus position by administering 8 mL of sterile PBS at 4 °C into the abdominal cavity. The abdominal region was softly massaged, and aspiration was performed using a needle coupled to a sterile syringe. The cells were counted in a Neubauer chamber by the Trypan Blue exclusion colorimetric method, and a minimum of 95% of living cells was obtained. The cells were counted using Neutral Red to obtain the desired concentration of macrophages ($2 \times 10^5$ cells/mL). Peritoneal macrophages were plated in each well, and incubated on the BCM. After 48 h of incubation at 37 °C and 5% $CO_2$, 10 µL of stained zymosan solution was added, and incubation continued for 30 min at 37 °C. Following this, 100 µL of Baker's fixative was added to paralyze the phagocytic process, and after 30 min the plate was washed with 0.9% saline solution to remove the zymosan and Neutral Red that were not phagocytized by macrophages. The supernatant was removed, 100 µL of extraction solution was added, and after solubilization on a Kline shaker, absorbance was measured at 550 nm in a BioTek plate reader (model ELx800) (*Souza et al., 2017*).

## Toxicity

To assess toxicity, the nitric oxide (NO) induction test was performed. Peritoneal macrophages ($2 \times 10^5$ per well) were plated and incubated with the BCM after 24 h of incubation at 37 °C and 5% $CO_2$. Cell supernatants were transferred to another 96-well plate for nitrite dosing. The standard curve was prepared with sodium nitrite diluted in Milli-Q water at 1, 5, 10, 25, 50, 75, 100, and 150 µM in the appropriate culture medium. At the different timepoints, the standard curve was determined with the same volume of Griess reagent (1% sulfanilamide in 10% $H_3PO_4$ [v:v] in Milli-Q water, added in equal parts to 0.1% naphthylenediamine in Milli-Q water), and the absorbance was read on a BioTek plate reader (model ELx800) at 550 nm. Lipopolysaccharide (LPS) was used as a positive control (*Sundaram et al., 2016*).

### BCM cytotoxicity

The basis of cytotoxicity assays is the evaluation of biomaterial-induced interference in cellular metabolic processes, and the investigation of processes that may intervene in cell growth/multiplication, or even culminate in cell death (*Ávila et al., 2014*).

According to *Boersema et al. (2016)*, cytotoxicity can be evaluated by different methods according to the type of cell damage: alterations in plasma membranes can be evaluated by means of dyes such as Trypan Blue and alamarBlue; alterations in the metabolic functions of mitochondria can be measured by the MTT (3-(4,5-dimethylthiazol-2-yl)-2,5-diphenyl tetrazolium bromide) colorimetric method.

The experiments were performed separately in 24-well plates. In the first plate, $2 \times 10^5$ macrophages per well were plated, and 500 µL of supplemented RPMI 1640 medium was added. In the second plate, $1 \times 10^5$ BM-MSCs in low-glucose DMEM were added. The plates were incubated at 37 °C and 5% $CO_2$ for 4 h to allow for cell adhesion. Two washes were performed with their respective media for removal of nonadherent cells. Subsequently, 500 µL of each medium was added, and the BCM (diameter 15.4 mm) was added. Macrophages were incubated for 48 h, and BM-MSCs for 7 days, followed by the addition of 10% 5 mg/mL MTT (diluted in medium). The macrophages and BM-MSCs were incubated for another 4 h in an incubator at 37 °C with 5% $CO_2$. The supernatant was discarded, and 100 µL of dimethyl sulfoxide (DMSO) was added to all wells. The BCM was removed, and the plate was shaken for 30 min on a Kline shaker (model AK 0506) at room temperature for complete dissolution of the formazan. The colorimetric reading was performed in a spectrophotometer at 550 nm in a BioTek plate reader (model ELx800). In the control group, the same conditions were applied to the culture media and the respective cultured cells (*Barud et al., 2015*).

### Statistical analysis

For analysis of phagocytic capacity, Student's *t*-test was used for independent samples of the cytotoxicity (MTT) and NO induction assays. GraphPad Prism version 5.0 was used to generate the graphs. These tests were performed in triplicate.

## RESULTS

Immediately after isolation, cells from the BM appeared rounded and dispersed, and floated in the culture medium. From the first day of culture, it was possible to identify undifferentiated cells with a fibroblastoid morphology that had adhered to the plastic. On day 2, cells appeared to still be in the adhesion process (Fig. 1A). The formation and proliferation of fibroblast colonies were evident on day 5 of culture. Colonies were of varying sizes, surrounded by empty spaces, and distributed throughout the culture plate. The cells showed well-defined cytoplasmic boundaries, and nuclei with regions of condensed chromatin; the closer they were to one another, the more elongated the cells were, and they were arranged parallel to one another (Fig. 1B).

In the observation on day 10, the cells adhered and arranged in colonies with 80% confluency in a 12-well plate (Fig. 1C). After the first passage, cells reached confluency

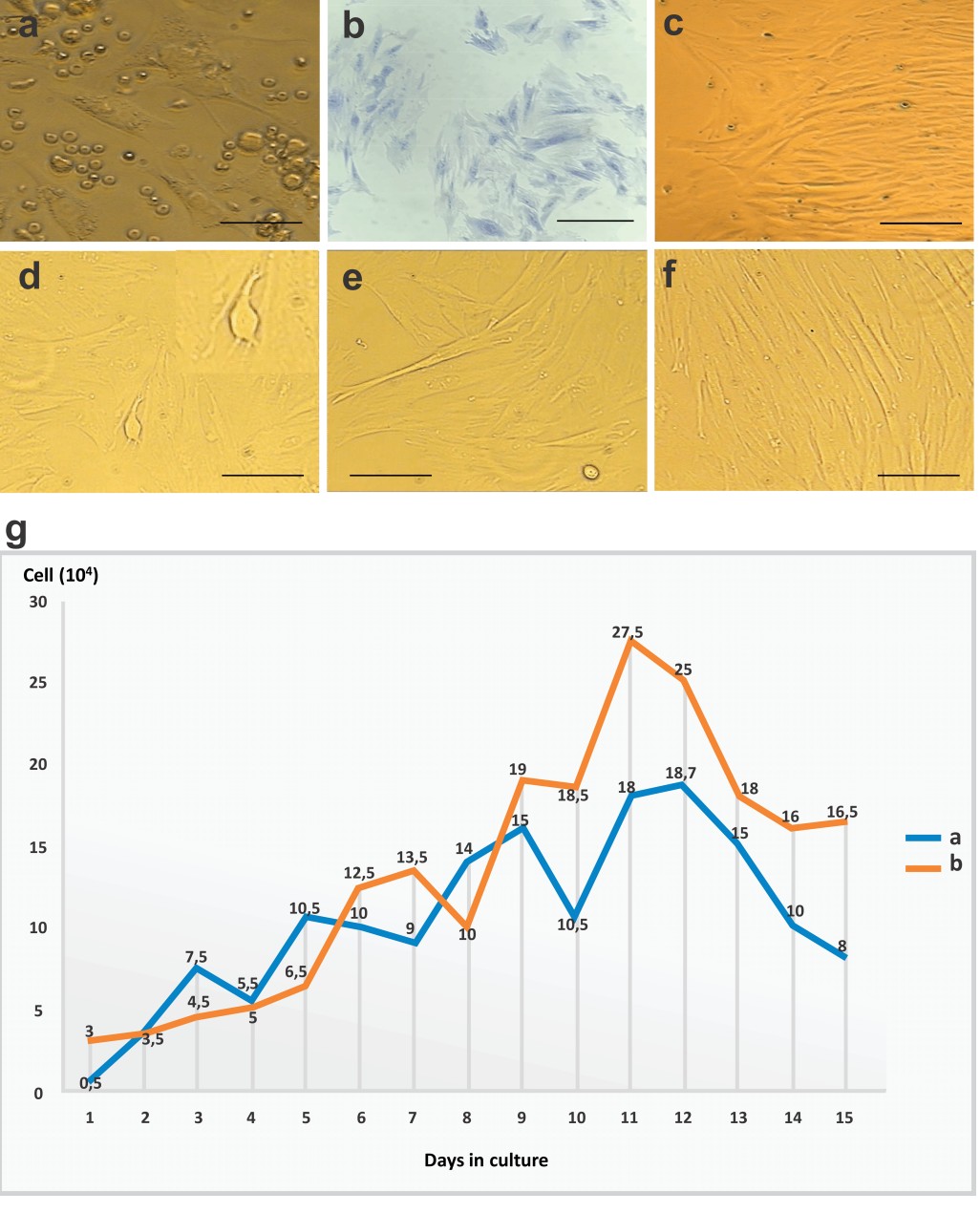

**Figure 1** **CFU-F assay, bone marrow mesenchymal stem cell (BM-MSC) in culture and growth curve of stem cells.** (A) Cells in the adhesion process on day 2 of cell culture performed in 12-well plates (objective 20×, bar: 25 μm). (B) CFU-F assay in a 24-well plate: photomicrography of Giemsa-stained BM-MSC colonies after 5 days of cell culture at 80% confluency, and colonies with more than 30 cells per field (objective 20×, bar: 25 μm), (C) cells arranged in parallel with fibroblastoid morphology at 80% confluency on day 10 of cell culture in 12-well plates (objective 10×, bar: 50 μm), (D) and (E) cytoplasmic adhesion and expansion with 80% confluency in 25 cm² bottles after trypsinization on day 15 of culture (objective 10×, bar: 50 μm), (F) cells with fibroblastoid morphology arranged in parallel and in colonies at 80% confluency in 25 cm² bottles after trypsinization on day 20 of culture (10× objective, bar: 50 μm) and (G) growth curve of stem cells derived from rabbit bone marrow during 15 days of culture after thawing, at a concentration of 1 ×10⁴ cells/mL. Phases identified: lag (days 1–4), log (days 5–11), and culture decline (days 12–15).

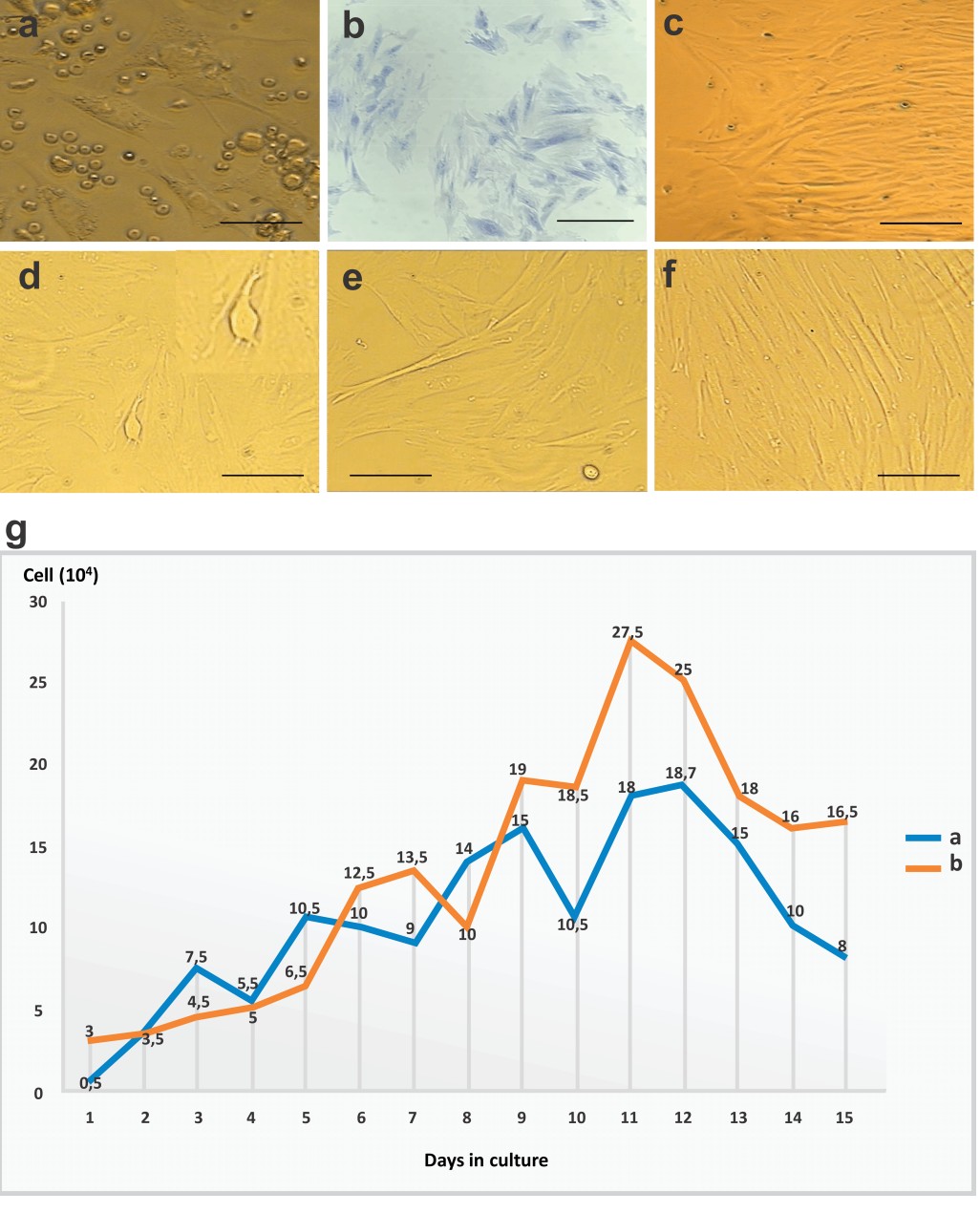

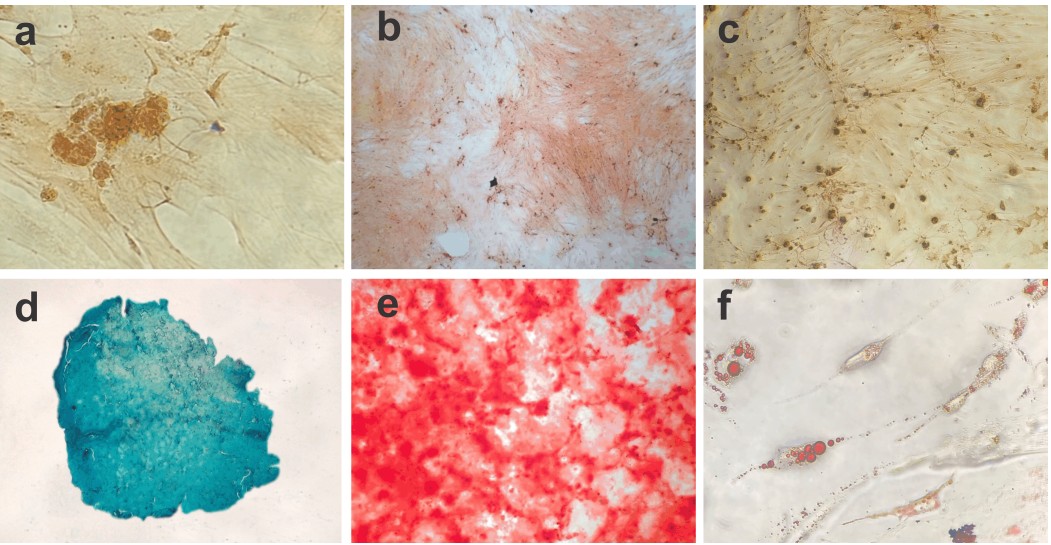

**Figure 2** **Photomicrographs showing BM-MSC differentiation.** (A) Negative control for 14 days of chondrogenic differentiation (objective 10×, bar: 25 μm), (B) negative control for osteogenic differentiation for 21 days (objective 10×, bar: 25 μm), (C) negative control for adipogênica differentiation for 10 days (objective 10×, bar: 25 μm), (D) BM-MSC chondrogenic differentiation (objective 20×, bar: 25 μm), (E) BM-MSC osteogenic differentiation showing calcium deposits in the extracellular matrix (objective 10×, bar: 25 μm) and (F) photomicrograph showing the adipogenic differentiation of BM-MSCs, with lipid vacuoles present in the cytoplasm stained red with Oil Red (objective 40×, bar: 25 μm).

more rapidly, with only a 5-day interval until 80% confluency was reached in 25 cm$^2$ bottles (Figs. 1D–1F).

After thawing, cell cultures exhibited viability of 96%, with similar morphological characteristics and maintenance of differentiation as the primary culture. The observed time to confluency was superior to that of the first passage of the primary culture. At day 3, the culture showed 80% confluency. In the growth curve, we identified two phases (lag and log) which corresponded to the adaptation period of the cells to the culture conditions, the exponential growth period, and the stability period with a reduction in cell growth. Data regarding cell concentration were used to evaluate cell kinetics, and are presented in Fig. 1G.

## Differentiation into BM-MSC mesodermal lineages

The cell differentiation assay showed the potential of BM-MSCs to differentiate into chondrogenic, osteogenic and adipogenic lineages. Following chondrogenic differentiation, cells were stained vibrant blue by Alcian Blue, and control cells presented some spontaneous differentiation. During osteogenic induction, the culture demonstrated increased deposition of calcium in the extracellular matrix from day 13 of culture. On day 21 of induction, the culture exhibited osteogenic characteristics, which were confirmed with Alizarin Red staining. The negative control showed adhered cells with morphology indicative of spontaneous differentiation foci. During adipogenic differentiation, the cells gradually changed to a fibroblastoid morphology, and the cytoplasmic lipid vacuoles became bulky (Fig. 2).

## BM-MSC biointegration with the BCM

In the BCM-associated cell culture, BM-MSCs with a fibroblastoid shape integrated with the biomaterial, and proliferation of the colonies was evident at 14 days of culture (Figs. 3A, 3B).

Using SEM, it was possible to observe that the rounded shape of the cells after 24 h of culture was maintained after being subtly anchored to the randomly arranged fibers of the BCM. After 7 days of culture, the cells presented themselves in groups, forming colonies with several fixation points, generating greater adhesion to the biomaterial. Micrographs recorded after 14 days of cell culture show BM-MSCs with their cytoplasm fully adhered to the BCM (Fig. 3).

## Macrophage activation and BCM cytotoxicity

In the phagocytic activity assay, Student's $t$-test was performed to determine the difference between the absorbance resulting from the association of macrophages with cellulose, and the control group (macrophages in the presence of 0.2% DMSO in RPMI 1640 medium). In the presence of the BCM, macrophage activity was significantly increased (Fig. 4A).

The colorimetric reading of NO release showed that the levels remained at a non-cytotoxic concentration for the cells in the presence of the BCM (Fig. 4B). The difference in NO release between the control and BCM was statistically significant at $p < 0.05$ ($p$-value 0.0184, $t_{0.05}$-critical: 2.6252), as was that between LPS and BCM ($p$-value: 0.0001; $t_{0.05}$-critical: 11.1963).

The tetrazole salt (MTT), incubated with cells with full metabolic activity, showed intense mitochondrial activity (Figs. 4C, 4D). In this trial, the metabolism of MTT by BM-MSCs showed a statistically significant difference ($p$-value: 0.0001; $t_{0.05}$-critical: 2.6252) but there was no statistically significant difference ($p$-value: 0.0628; $t_{0.05}$-critical: 2,000) between the BCM associated with murine macrophages and with the control. In both conditions, cell viability was greater than 94% (Figs. 4E, 4F).

## DISCUSSION

After isolation, BM-MSCs exhibited a rounded shape in culture. During the adhesion and expansion process, their morphology modified, becoming gradually fusiform, and proliferating in parallel in colonies; the exclusion of hematopoietic cells in the medium exchanges was perceptible. Similarly, *Zhang et al. (2014)* stated that MSCs adhere to favorable surfaces with rapid morphological changes, ranging from rounded to elongated shapes. According to *Ikebe & Suzuki (2014)*, adhesion to plastic is the first criterion for the characterization of MSCs. In the cellular adhesion phase, physicochemical connections occur between the cells and the contact surface, including ionic forces that rapidly alter cell morphology, and which are evident after 1 h of culture (*Bakhtina et al., 2014*; *PU & Komvopoulos, 2014*; *Wang et al., 2016*).

The organization of cells in fibroblastoid colonies has been considered by *Kisiel et al. (2012)* as the second major characteristic of MSCs. In this experiment, colony formation was evident after 5 days of primary culture, suggesting that these interactions can occur without cellular differentiation, and therefore allow fibroblastoid morphology to be maintained.
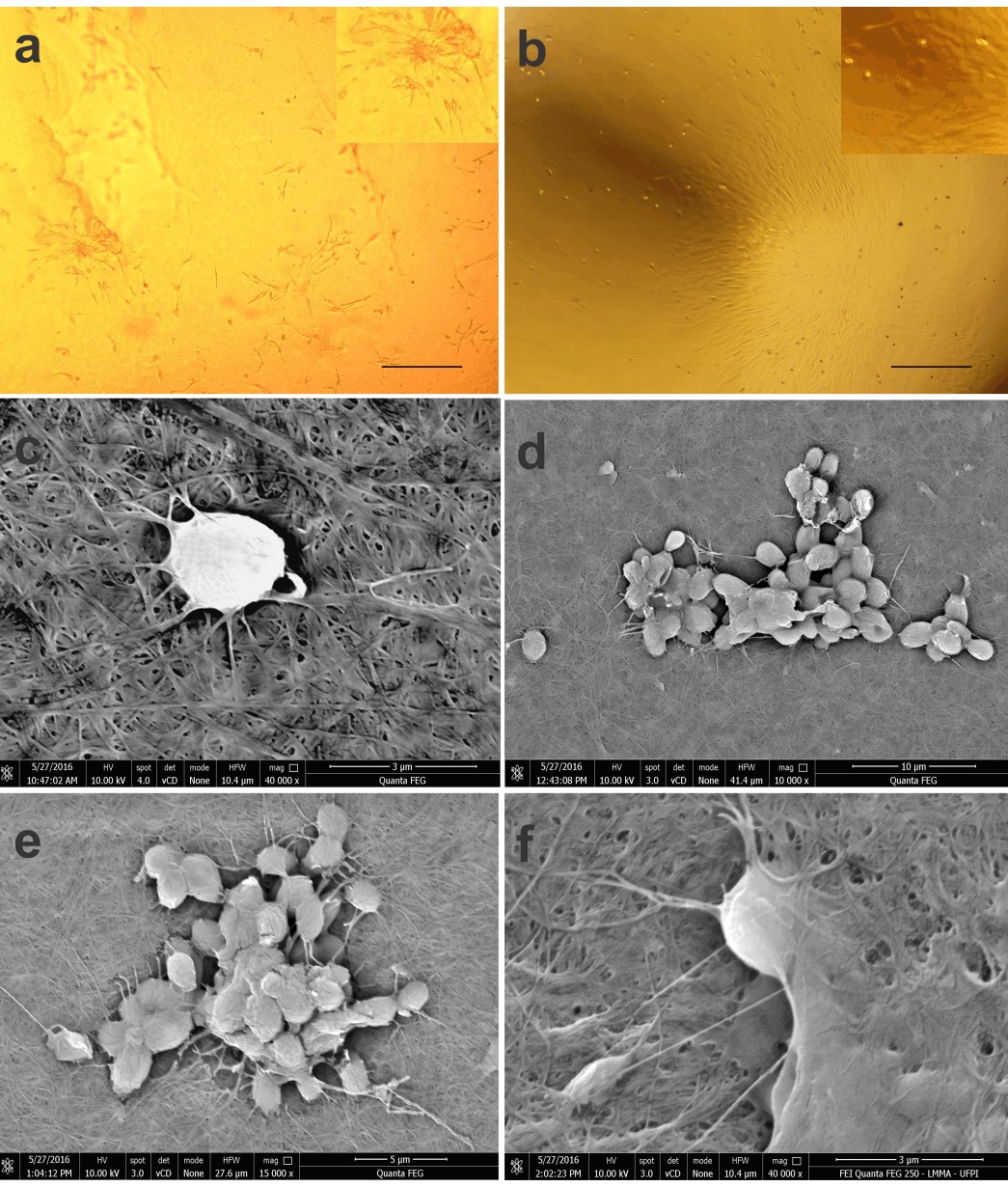

**Figure 3** **Photomicrographs of BM-MSCs adhered to the bacterial cellulose membrane (BCM) and scanning electron microscopy showing BM-MSC anchorage and biointegration with the BCM.** (A) BM-MSC adhesion after 7 days of cell culture, highlighting the formation of CFU-F on the BCM (objective 20×, bar: 25 μm), (B) BM-MSC colonies after 14 days of culture (objective 10×, bar: 50 μm), (C) analysis after 24 h of cell culture (40,000×), (D) and (E) with after 7 (10,000× and 15,000× respectively) and (F) 14 days of culture (40,000×).

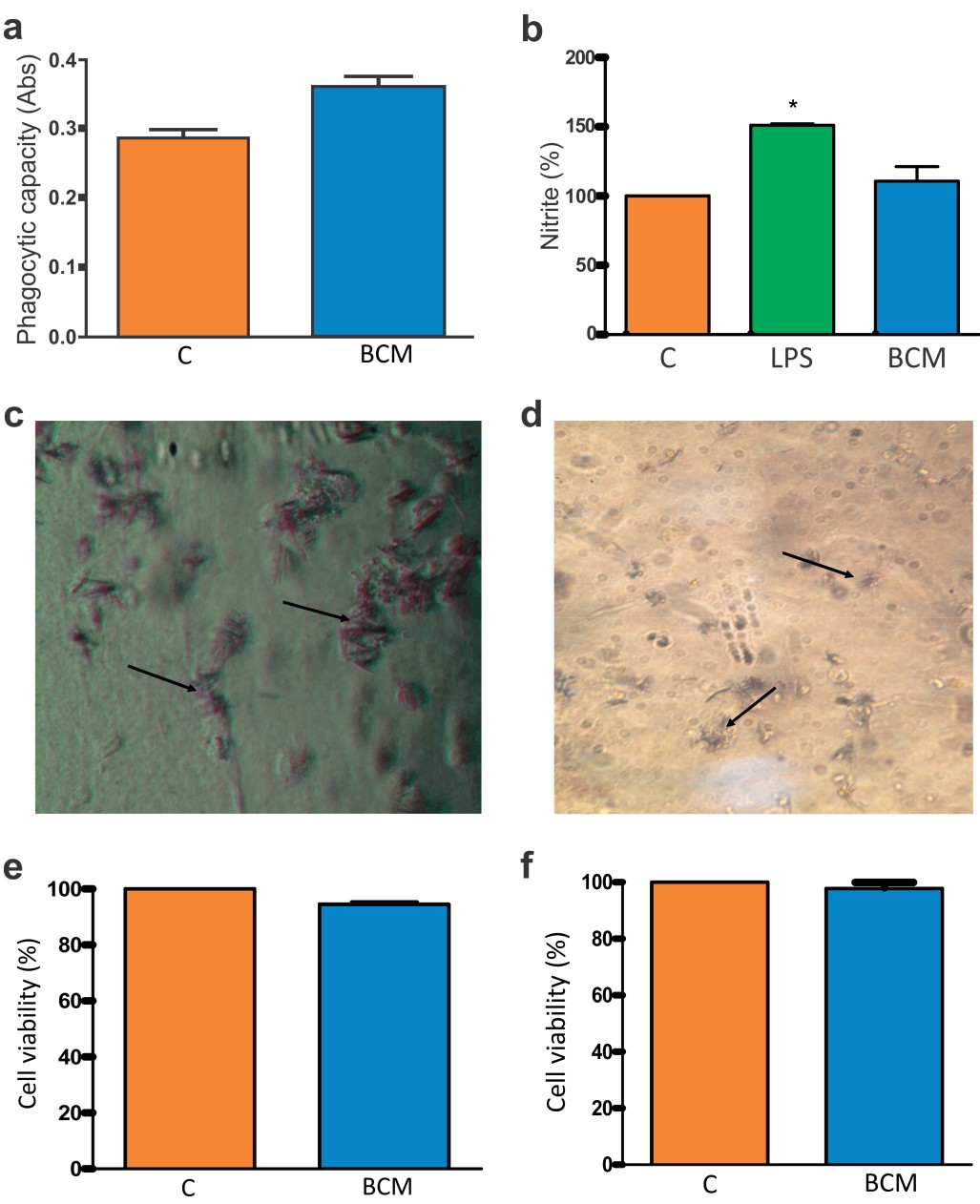

**Figure 4** **Macrophage activation and BCM cytotoxicity.** (A) Zymosan particle phagocytosis by macrophages in the presence of the BCM. The graph represents the mean ± standard error of the mean of three independent experiments performed in triplicate (control: mean 0.28567, standard deviation 0.03161; BCM, mean 0.36100, standard deviation 0.03474). ABS, absorbance; C, control; BCM, bacterial cellulose membrane; $*p < 0.05$. (B) Colorimetric nitrite dosage produced by macrophages treated with lipopolysaccharide (LPS) in the presence of the BCM. The plot represents the mean ± standard error of the average of three independent experiments performed in triplicate (control, mean 100.0000, standard deviation 0.0000; LPS, mean 150.8889, standard deviation 1.0541; BCM, mean 109.6300, standard deviation 11.0047). Student's $t$-test was performed for comparison between groups and the control (0.2% dimethyl sulfoxide [DMSO] in RPMI 1640 medium). 

**Figure 4 (…continued)**
C, control; LPS, lipopolysaccharide; BCM, bacterial cellulose membrane; $*p < 0.05$. (C) Formazan crystals in BCM cultured with peritoneal macrophages and (D) BM-MSCs. Increasing view 40×. (E) BM-MSC viability in the BCM (control, mean 100.0000, standard deviation 0.0000; BCM, mean 94.4533, standard deviation 1.1926), and (F) viability of murine macrophages in the BCM (control, mean 100.0000, standard deviation 0.0000; BCM, mean 97.7867, standard deviation 3.3200). The plot represents the mean ± standard error of the mean of three independent experiments performed in triplicate. Student's $t$-test was performed to compare the groups with the control (0.2% DMSO in DMEM/RPMI medium). C, control; BCM, bacterial cellulose membrane; $*p < 0.05$.

Regarding cell viability after thawing, the lag phase was evident from day 1 to day 4 of the growth curve, and the log phase occurred between days 5 and 11, with exponential mitotic divisions evident mainly between days 9 and 11; a decline in the number of cell divisions occurred between days 12 and 15. *Secunda et al. (2015)* defined the lag phase as a relatively short stage characterized by onset of the release of cell proliferation factors. The exponential cell growth (log) phase is the second phase, in which the growth rate and duration depend on the medium used. When cellular metabolism can no longer be maintained, cells undergo apoptosis. Levels of confluence above 90% induce cell death through a mechanism of inhibition by contact, triggering apoptosis, in addition to the reduction of substrate levels of the culture medium, due to the high cellular concentration, as described by *Meirelles & Nardi (2003)*.

The ability to differentiate into more than one mesenchymal lineage (chondrogenic, osteogenic, or adipogenic) is an important multipotentiality feature of MSCs, and is a fundamental requirement for their characterization (*Wuchter, wagner & Ho, 2016*). According to *Kolf et al. (2015)*, the tissue formed by chondrogenic cell differentiation acquires a vibrant blue color when stained with Alcian Blue; during osteogenic differentiation, it is possible to observe the gradual deposition of calcium in the extracellular matrix, which is attributable to the presence of osteoblasts. Alizarin Red staining showed a fairly characteristic reddish coloration, providing evidence of this potential. According to *Munir et al. (2017)*, formation of lipid vacuoles in the cell cytoplasm and staining by Oil Red characterize the formation of adipocytes; during adipogenic differentiation, several independent vacuoles can be found, and fuse as they expand inside the cell. In this study, cell culture using specific media for differentiation into mesodermal (chondrogenic, osteogenic, or adipogenic) lineages demonstrated the multipotentiality of rabbit BM-MSCs.

Cell adhesion and proliferation largely depend on the characteristics of the biomaterial surface, since interactions that occur on the surface will drive the biological responses (*Chahal et al., 2016*; *Khayyeri et al., 2016*). After 7 days of culture, cells showed organization in a fibroblastoid format with a tendency for cell grouping. In the analysis performed at 14 days, the BM-MSCs were present in colonies, and covered the BCM surface.

Using SEM, we verified that BM cells maintain their rounded shape on the BCM surface in the first 24 h, with few biomaterial fixation bridges. A delay in BM-MSC anchoring to the BCM was observed when compared to adhesion in culture plates, and this anchoring onset was evident in a few hours. According to *Silveira et al. (2016)*, the three-dimensional structure of BCM nanofibers exhibits an arrangement similar to that of the collagen fibers

of the extracellular matrix, and a surface with different pores can provide variable times for cell adhesion to the biomaterial.

BM-MSC anchoring and proliferation on the BCM were evident on day 7 of culture with grouped cells, and several cytoplasmic projections were evident in the BCM. On the 14th day of culture, fixation of the BM-MSCs occurred by interaction with the biomaterial. Consistent with the studies of *Alberti & Xu (2016)* and *Santana, Neto & Sá (2014)*, the presence of cytoplasmic projections and normal cell morphology are factors that confirm cytocompatibility between the BCM scaffold and cells.

Equilibrium in immune system cell activation also reflects a tissue's regenerative quality. In the presence of the BCM, the macrophages presented a statistically significant increase (*p*-value 0.0002; $t_{0.05}$-critical: 4.8118) in their activity compared with the control group. *Qiu et al. (2016)* clarified that maintaining the scaffold intact during the period of adhesion and cell proliferation is important for the regenerative process and the architecture of the tissue to be repaired. The implanted biomaterial should gradually biodegrade to give rise to newly formed tissue without exacerbating an inflammatory response that compromises the repair quality. Thus, the adequate inflammatory response of the host in specific situations makes the biomaterial compatible with its use.

The ability of bacterial cellulose to be degraded has not yet been fully elucidated. In animal and human tissues, it is considered limited due to the absence of hydrolases that rupture the ß (1,4) binding of the cellulose chain, which is responsible for the solubility of the biomaterial (*Oliveira, Rambo & Porto, 2013*). Although the idea of a completely degradable scaffold is interesting from the point of view of tissue engineering, there remain difficulties with materials that exhibit this property, since the timing of degradation and tissue repair combined with the mechanical properties acquired by the newly formed tissue have led researchers to believe that a material with a low rate of degradation may respond better when the scarring process requires more time-consuming conditions (*Bhattacharjee et al., 2015*).

After inflammation, macrophages release NO as a way to eliminate pathogens. In addition, NO is known as an inflammatory response mediator, inhibiting or inducing inflammation according to the concentration of NO released (*Taraballi et al., 2016*). The colorimetric nitrite dosage produced by macrophages in the presence of the BCM showed a non-cytotoxic concentration, approaching the value obtained in the control group.

The MTT assay is a method to assess cell viability widely used to evaluate the metabolism of MTT in the mitochondria of viable cells when incubated with cells with full metabolic activity crossing the plasma membrane, and which, when coming in contact with the superoxide produced by mitochondrial activity, is reduced by succinate dehydrogenase present in MTT-formazan-containing mitochondria. The crystals formed are insoluble in water; however, they are solubilized in DMSO medium, and show violet coloration. Thus, cell viability is directly proportional to the intensity of staining (*Toh, Yap & Lim, 2015*). According to *Li, Zhou & Xu (2015)*, a material is considered non-cytotoxic and biocompatible when cell viability is greater than 70%. In this study, the MTT assay presented intense violet staining, showing that the BCM does not produce a toxic effect

on the cells; 94% cellular viability is considerably favorable for non-interference of cellular activity.

## CONCLUSION

The expansion and cellular integration of biomaterials depends greatly on the quality and suitability of the biomaterial surface. The BCM allowed the adhesion, expansion, and biointegration of BM-MSCs, and the cytotoxicity and toxicity of the BCM were low enough to maintain considerable viability in cell culture. Macrophage activation and the rate of BCM degradation make the BCM an ideal biomaterial for slow healing processes in which reconstructed tissues require a scaffold with longer durability.

Considering the interaction demonstrated between BM-MSCs and the BCM, it can be stated that the BCM is a promising biomaterial in tissue engineering and regenerative medicine. However, it will be necessary to test the behavior of BCM implants *in vivo*.

## ACKNOWLEDGEMENTS

We thank the Integrated Nucleus of Morphology and Research with Stem Cells and the laboratories: Interdisciplinary of Advanced Materials, Advanced Microscopy Multiuser and Antileishmania Activity, Federal University of Piauí—UFPI, besides the Laboratories of Toxicological Research and Industrial Microbiology and Process of Fermentation of the University of Sorocaba—UNISO.

### Funding

This work had the financial support of the National Council of Scientific and Technological Development-CNPq (Process: 427626/2016-1). The funders had no role in study design, data collection and analysis, decision to publish, or preparation of the manuscript.

### Grant Disclosures

The following grant information was disclosed by the authors:
National Council of Scientific and Technological Development-CNPq: 427626/2016-1.

### Competing Interests

The authors declare there are no competing interests.

### Author Contributions

- Marcello de Alencar Silva and Fernando Aécio de Amorim Carvalho conceived and designed the experiments, performed the experiments, analyzed the data, contributed reagents/materials/analysis tools, prepared figures and/or tables, authored or reviewed drafts of the paper, approved the final draft.
- Yulla Klinger de Carvalho Leite, Camila Ernanda Sousa de Carvalho and Michel Muálem de Moraes Alves performed the experiments, analyzed the data, authored or reviewed drafts of the paper, approved the final draft.

- Matheus Levi Tajra Feitosa conceived and designed the experiments, authored or reviewed drafts of the paper, approved the final draft.
- Bartolomeu Cruz Viana Neto performed the experiments, contributed reagents/materials/analysis tools, authored or reviewed drafts of the paper, approved the final draft.
- Maria Angélica Miglino conceived and designed the experiments, performed the experiments, analyzed the data, prepared figures and/or tables, authored or reviewed drafts of the paper, approved the final draft.
- Angela Faustino Jozala conceived and designed the experiments, analyzed the data, authored or reviewed drafts of the paper, approved the final draft.
- Maria Acelina Martins de Carvalho conceived and designed the experiments, analyzed the data, prepared figures and/or tables, authored or reviewed drafts of the paper, approved the final draft.

### Animal Ethics

The following information was supplied relating to ethical approvals (i.e., approving body and any reference numbers):

The study was carried out in accordance with the recommendations of the Guide for the Laboratory Animals Care and Use of the National Institute of Health. The protocol was approved by the Ethics Committee on Animal Use of the Federal University of Piauí (CEUA-UFPI, permit number: 268/16).

### Data Availability

The raw data are provided in Supplemental Information 1.

### Supplemental Information

Supplemental information for this article can be found online at http://dx.doi.org/10.7717/peerj.4656#supplemental-information.

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
