# Peer review of "Behavior and biocompatibility of rabbit bone marrow mesenchymal stem cells with bacterial cellulose membrane"

_PeerJ, doi:10.7717/peerj.4656_

## Round 0.1 · original submission · Major Revisions

Please respond in detail to the comments of the two reviewers. Note that Reviewer 1 has provided an additional document

Reviewer 1 ·

Basic reporting

Style of reporting is inconsistent.
Language needs to be improved.
e.g. 56 Researchers have been studying bone marrow mesenchymal stem cells, the cells
57 applicability in regenerative medicine and improving methodologies presented a greater
58 effectiveness (Dimarino, Caplan & Bonfield, 2013; Wei et al., 2013; Li et al., 2016).
59 The mesenchymal stem cells use appears promising in the regenerative medicine field.
60 On the other hands, these studies developed,
66 The tissue engineering comprises a promising multidisciplinary field that involves
67 materials development or devices capable of specific interactions with biological tissues (Langer
68 & Vacanti, 2016). The researches expansion in this area has accentuated the search for subsidies
69 to suport the stem cells biocompatibility in biopolymers to attend in vitro tissues develops to
70 replace injured areas.
71 In the last years
83 In tissue engineering the bacterial cellulose supported the researches interest because it is
84 an abundant biopolymer in nature with biodegradability and able to be synthesized by several
85 bacteria adding the low cost in its manufacture. The nanofibrils formed between 20-100nm.
86 Their which intertwine forming a network and allied to its molecular structure, giving its high
87 hydrophilicity. Many biomedical researches have
101 A one year old male New Zealand rabbit considered clinically healthy, was used for the
102 isolation procedure of BMMSC. A mouse Mus muscles was used as a peritoneal machophages
103 source. For the cellular viability determination trypan blue and subsequent graphical
104 representation in the growth curve were done. For the fibrolastoid colony forming units (CFU-F)
105 assay were used cells collected from the bone marrow cultured in petri dish in the sixth passage.
106 For the differentiation potential in mesenchymal lineages study were used the means of
107 chondrogenic and osteogenic inductionic. In the verification of BMMSC biointegration to BCM
108 confocal microscopy and Scanning Electron Microscopy (SEM) were used and to analyze the
109 BCM phagocytic capacity, toxicity and cytotoxicity were used peritoneal macrophages
sowing the
164 concentration of 1 x 104 cells/mL in five six-well plates counting two wells every 24 hours in the.

Recent references on in vivo studies using bacterial cellulose based scaffolds need to be included.
The results are relevant to hypothesis

Experimental design

Methods have been described in sufficient detail.

Validity of the findings

The data on differentiation of the stem cells needs to be supported by atleast one more independent assay/technique

Additional comments

Major inconsistencies are observed with reference to citing the references.
Several typographic errors are observed.
Improvement in the language of presentation of paper is suggested.
Image quality (light microscopy images) need to be better.
An annotated PDF is being provided. Most of the suggested corrections have been marked.

Annotated reviews are not available for download in order to protect the identity of reviewers who chose to remain anonymous.

Reviewer 2 ·

Basic reporting

The paper would be significantly improved by careful editing of the English language and grammer. It is hard to understand in places. Some phrases are misleading (eg homogenised instead of mixed, peal instead of passage in differentiation instead of undifferentiated-line 269). There are also typographical errors which should be fixed
The way the paper is structured leaves me unclear about the author’s aims. The initial focus is on the isolation of rabbit-derived bone marrow stromal cells, these are then cultured on a bacterial cellulosic membrane. The paper then switches to some characterisation of the membrane. It might be clearer to describe the membrane properties then to present the cell isolation data then to assess the cell and membrane combination.
In the discussion light and shading are included as physical conditions which influence mammalian cell culture, (lines 341 and 343) can the authors support this with references? I don’t believe this to be a factor for stromal cell culture.
The references cited are very heavily biased towards the last 4-5 years. The methods section should include citations to relevant papers, particularly around the cell isolation and characterisation used.
Some figures are not very clear. In figure 4 it is hard to see any cells. Can the stated magnification for figures 8 C and D be checked?

Experimental design

The aims and objectives of the work could be more clearly stated in the introduction.
Citations should be added to the methods.
The experimental methods are not clear in several places
-Was only 1 rabbit used to isolate MSCs? How reproducible was the cell isolation and characterisation-including the behaviour of the cells on the bacterial cellulosic membrane?
-Was the chondrogenesis assay conducted on a monolayer of cells? (Normally these assays are done in a pellet or micromass culture). Can the authors comment on which protocol was used and cite the appropriate paper?
-At what stage was the CFU-F assay performed? On cells directly isolated from marrow or after the MSCs had been in culture for some time? If the former do the authors have data on the number of CFUFs isolated as a function of the number of cells plated?
-Can the detail about the number of peritoneal macrophages added to the bacterial cellulosic membrane be added?
-How was the BCM isolated and prepared? How were MSCs seeded onto this?

Validity of the findings

Data are presented showing isolation of MSCs from rabbit marrow. This seems to be from 1 animal. How reproducible is this work-in particular in relation to the behaviour of the cells on BCM?
Chondrogenesis data are shown based on what appears to be a monolayer assay which is unconventional. The staining shown is not of high quality-the sample appears to have peeled and folded onto itself. What do the authors mean by “cytoplasmic integration” in line 287?
There is a confusing jump in the data from rabbit MSCs on BCM to characterisation of BCM which should be better explained in the results.
Figure 9 shows that the BCM causes a significant increase in macrophage activity- what does this mean? Some explanation would be helpful.
The authors describe the levels of NO production as non-cytotoxic (line 310) can they put this into context?

Part of the discussion is a validation of the rabbit MSCs by comparison to definitions of MSCs from different groups, the authors should compare their cells to others who have isolated MSCs from rabbit bone marrow. There are publications from over a decade ago which isolated these cells.

---

## Round 0.2 · Minor Revisions

The authors have significantly improved the quality of the manuscript, however, some more changes may further improve the manuscript:

in the response letter to the reviewers critiques, it not clear at many places what changes have been made to the manuscript. For example the reviewer suggested that citations should be added to the methods and the reply of the authors was "the suggestions were followed" but not giving more detail.

I encourage the authors to resubmit a detailed point to point reply detailing the changes introduced in the revision and also giving the line/page number.

The authors should try to combine some of the figures. In my opinion none on the figures merits stand alone status. Currently the authors have 12 figures, this should be significantly reduced.

---

## Round 0.3 · accepted · Accept

Dear author,

Thank you for improving the manuscript.

I am pleased to inform you that it is now accepted for publication in PeerJ
Regards

#